# Blood pressure-lowering treatment for the prevention of cardiovascular events in patients with atrial fibrillation: An individual participant data meta-analysis

Ana-Catarina Pinho-Gomes[1], Luis Azevedo[2], Emma Copland[3], Dexter Canoy[3,4], Milad Nazarzadeh[3], Rema Ramakrishnan[3], Eivind Berge[5,6†], Johan Sundström[7], Dipak Kotecha[8], Mark Woodward[9,10,11], Koon Teo[12], Barry R. Davis[13], John Chalmers[9], Carl J. Pepine[14], Kazem Rahimi[3,4]*, on behalf of the Blood Pressure Lowering Treatment Trialists' Collaboration¶

1 Faculty of Life Sciences and Medicine, King's College London, London, United Kingdom, 2 Department of Community Medicine, Information and Health Decision Sciences, Centre for Health Technology and Services Research, Faculty of Medicine, University of Porto, Porto, Portugal, 3 Deep Medicine, Nuffield Department of Women's and Reproductive Health, University of Oxford, Oxford, United Kingdom, 4 National Institute for Health Research Oxford Biomedical Research Centre, Oxford University Hospitals National Health Service Foundation Trust, Oxford, United Kingdom, 5 Department of Cardiology, Oslo University Hospital, Oslo, Norway, 6 Institute for Clinical Medicine, University of Tromsø, Norway, 7 Department of Medical Sciences, Uppsala University, Sweden, 8 Institute of Cardiovascular Sciences, University of Birmingham, Birmingham, United Kingdom, 9 The George Institute for Global Health, University of New South Wales, Sydney, Australia, 10 The George Institute for Global Health, Department of Epidemiology and Biostatistics, Imperial College, London, United Kingdom, 11 Department of Epidemiology, Johns Hopkins University, Baltimore, Maryland, United States of America, 12 Population Health Research Institute, McMaster University, Hamilton, Ontario, Canada, 13 The University of Texas School of Public Health, Houston, Texas, United States of America, 14 Department of Medicine, University of Florida, Gainesville, Florida, United States of America

† Deceased.
¶ Writing group, steering committee, and collaborators are listed in the Acknowledgments.
* kazem.rahimi@wrh.ox.ac.uk

**Data Availability Statement:** The governance of BPLTTC and policies on data access and sharing policies are described elsewhere (Rahimi K, et al.

## Abstract

### Background

Randomised evidence on the efficacy of blood pressure (BP)-lowering treatment to reduce cardiovascular risk in patients with atrial fibrillation (AF) is limited. Therefore, this study aimed to compare the effects of BP-lowering drugs in patients with and without AF at baseline.

### Methods and findings

The study was based on the resource provided by the Blood Pressure Lowering Treatment Trialists' Collaboration (BPLTTC), in which individual participant data (IPD) were extracted from trials with over 1,000 patient-years of follow-up in each arm, and that had randomly assigned patients to different classes of BP-lowering drugs, BP-lowering drugs versus placebo, or more versus less intensive BP-lowering regimens. For this study, only trials that had collected information on AF status at baseline were included. The effects of BP-lowering treatment on a composite endpoint of major cardiovascular events (stroke, ischaemic heart

BMJ Open 2019;9:e028698). Our data sharing agreements with our collaborators limit us from sharing the original data to third parties. However, a governance framework exists for collaborative projects with external research investigators. For further queries, please check www.bplttc.org for our contact details.

**Funding:** The following authors are supported by grants from the British Heart Foundation "ACPG, KR & DC (grant number: FS/19/64/34673), KR & DC (grant number: PG/18/65/33872), MN, KR & DC (grant number FS/19/36/34346); KR is also in receipt of funding from the UKRI's Global Challenges Research Fund (Grant Ref ES/P011055/1), the Oxford NIHR Biomedical Research Centre, and the Oxford Martin School at University of Oxford. The funders had no role in study design, data collection and analysis, decision to publish, or preparation of the manuscript. The views expressed are those of the authors and not necessarily those of the National Health Service, the NIHR or the Department of Health and Social Care. This manuscript was prepared using ACCORD, ALLHAT and SHEP Research Materials obtained from the NHLBI Biologic Specimen and Data Repository Information Coordinating Centre and does not necessarily reflect the opinions or views of the ACCORD, ALLHAT and SHEP, or the NHLBI.

**Competing interests:** I have read the journal's policy and the authors of this manuscript have the following competing interests: KR is an academic editor on PLOS Medicine's editorial board and has in the past received personal fees as Speciality Consulting Editor for PLOS Medicine. KR is also in receipt of personal fees as Associate Editor for BMJ Heart.MW reports personal fees from Amgen, Kyowa Kirin, and Freeline outside the submitted work; JS reports ownership in companies providing services to Itrim, Amgen, Janssen, Novo Nordisk, Eli Lilly, Boehringer, Bayer, Pfizer and AstraZeneca, outside the submitted work. Group authors listed in the acknowledgements have no known competing interests.

**Abbreviations:** AF, atrial fibrillation; BP, blood pressure; BPLTTC, Blood Pressure Lowering Treatment Trialists' Collaboration; CCB, calcium channel blockers; CI, confidence interval; DBP, diastolic BP; HF, heart failure; HR, hazard ratio; IPD, individual participant data; PRISMA, Preferred Reporting Items for Systematic Reviews and Meta-Analyses; RAAS, renin-angiotensin-aldosterone system; RCT, randomised controlled trial; SBP, systolic BP.

disease or heart failure) according to AF status at baseline were estimated using fixed-effect one-stage IPD meta-analyses based on Cox proportional hazards models stratified by trial. Furthermore, to assess whether the associations between the intensity of BP reduction and cardiovascular outcomes are similar in those with and without AF at baseline, we used a meta-regression. From the full BPLTTC database, 28 trials (145,653 participants) were excluded because AF status at baseline was uncertain or unavailable. A total of 22 trials were included with 188,570 patients, of whom 13,266 (7%) had AF at baseline. Risk of bias assessment showed that 20 trials were at low risk of bias and 2 trials at moderate risk. Meta-regression showed that relative risk reductions were proportional to trial-level intensity of BP lowering in patients with and without AF at baseline. Over 4.5 years of median follow-up, a 5-mm Hg systolic BP (SBP) reduction lowered the risk of major cardiovascular events both in patients with AF (hazard ratio [HR] 0.91, 95% confidence interval [CI] 0.83 to 1.00) and in patients without AF at baseline (HR 0.91, 95% CI 0.88 to 0.93), with no difference between subgroups. There was no evidence for heterogeneity of treatment effects by baseline SBP or drug class in patients with AF at baseline. The findings of this study need to be interpreted in light of its potential limitations, such as the limited number of trials, limitation in ascertaining AF cases due to the nature of the arrhythmia and measuring BP in patients with AF.

## Conclusions

In this meta-analysis, we found that BP-lowering treatment reduces the risk of major cardiovascular events similarly in individuals with and without AF. Pharmacological BP lowering for prevention of cardiovascular events should be recommended in patients with AF.

## Author summary

### Why was this study done?

- Atrial fibrillation (AF) is the most common cardiac arrhythmia across the world and is strongly associated with future vascular disease, particularly stroke.

- Blood pressure (BP) lowering is an established strategy for prevention of vascular disease, but whether patients with AF benefit similarly from pharmacological BP reduction is not well understood.

### What did the researchers do and find?

- We compared the preventive effect of BP-lowering treatment on cardiovascular outcomes in patients with and without AF at baseline.

- We conducted an individual participant data meta-analysis using published and unpublished data from large randomised clinical trials (22 trials involving 188,570 patients).

- We showed that BP-lowering treatment reduced the risk of a major cardiovascular events with no evidence that effects differed according to the presence or absence of AF at baseline.

- The relative risk reductions were proportional to the intensity of BP reduction in individuals with and without AF.

- In individuals with AF, the relative risk reduction was comparable irrespective of whether baseline systolic BP was under or over the conventional treatment threshold of 140 mm Hg.

### What do these findings mean?

- BP-lowering treatment reduces the risk of major cardiovascular events in patients with AF to a similar extent to that of patients without AF.

- Pharmacological BP-lowering treatment for prevention of cardiovascular events should be recommended as part of care for patients with AF.

## Introduction

Atrial fibrillation (AF) is the most common clinically relevant cardiac arrhythmia and its incidence and prevalence are on the rise across the globe [1,2], mainly due to population ageing and an increase in other cardiometabolic risk factors [3]. In observational studies, AF has been associated with an approximately 90% higher risk of a fatal vascular event, such as stroke, ischaemic heart disease, heart failure (HF), and vascular dementia [4]. Although the risk of stroke, in particular, can be mitigated by anticoagulation, the majority of deaths in contemporary anticoagulated AF patients are due to cardiovascular causes other than stroke, such as myocardial infarction and HF [5,6]. Yet, there is no proven pharmacological intervention other than anticoagulation for effective reduction of such risks [7].

Although high blood pressure (BP) is the most common cardiovascular risk factor in patients with AF [8,9], whether BP lowering reduces the risk of cardiovascular events in patients with AF remains uncertain. As BP-lowering treatment significantly decreases cardiovascular risk in high-risk populations [10], a similar effect could be expected in patients with preexisting AF. However, the complex structural, neurohumoral, and metabolic changes in the cardiovascular system that underpin the development and progression of AF may interfere with BP-lowering treatment [11]. This uncertainty is further compounded by the fact that the only randomised controlled trial (RCT) specifically conducted in patients with AF failed to detect a risk reduction in cardiovascular events using an angiotensin receptor blocker [12]. Several other major BP-lowering trials have included patients with known AF, but the low prevalence of AF rendered them individually underpowered to perform subgroup analysis according to AF status at baseline. We have sought to extract previously published and unpublished data to compare the effect of BP-lowering treatment on fatal and nonfatal cardiovascular outcomes in patients with and without AF overall and by major drug classes.

## Methods

### Study design

We conducted individual participant data (IPD) meta-analyses of BP-lowering RCTs that investigated treatment effects on cardiovascular outcomes by presence or absence of AF at

randomisation. The study was based on the resource provided by the Blood Pressure Lowering Treatment Trialists' Collaboration (BPLTTC). RCTs are eligible for inclusion in BPLTTC if they have randomised participants to BP-lowering drugs versus placebo or alternative classes of BP-lowering drugs, or between more versus less intensive regimens, and have at least 1,000 patient-years of follow-up in each randomised arm. To date, 50 RCTs have shared data. Details of the methods underlying the latest cycle of the BPLTTC have been recently published and are described in S1 Methods [13]. A separate ethical approval was not required for this study. This analysis followed a prespecified protocol that is available as Supporting information (S1 Protocol).

In this study, only trials that had collected information on AF status at baseline were included. Three types of trials were identified: (1) trials that included both patients with and without AF at baseline; (2) trials that included only patients with AF at baseline; and (3) trials that excluded patients with AF at baseline. We excluded trials in which the presence of AF was not explicitly assessed at baseline or in which AF status at baseline was not clear.

## Definition of outcomes

The primary outcome was total cardiovascular events, defined as the first occurrence of (1) fatal or nonfatal stroke; (2) fatal or nonfatal myocardial infarction or ischaemic heart disease; or (3) HF causing death or requiring hospitalisation. Secondary outcomes were the individual elements of the composite endpoint as well as cardiovascular death and all-cause death.

## Treatment comparisons

For the main analysis, intervention and control groups were compared. For placebo-controlled trials, the placebo arm was considered as the "comparator" and the active treatment was considered as the "intervention." For trials with two or more active treatment arms, the arm in which the BP reduction was higher was considered as "intervention" and the other treatment arm(s) as "comparator." Treatment arms were grouped together whenever required to avoid double counting of participants. S1 Table summarises the treatment comparisons considered in each trial and the difference in systolic BP (SBP) reduction between trial arms.

## Risk of bias assessment

We used the Rob2 tool from the Cochrane Collaboration for assessing risk of bias of individual trials. (S2 Table) [14].

## Statistical analysis

Our main analyses aimed to address 4 questions: (q1) whether the effect of BP-lowering treatment on CVD outcomes differs between those with and without AF; (q2) whether the associations between the intensity of BP reduction and outcomes are similar in those with and without AF at baseline; (q3) whether in patients with AF, treatment effects vary by baseline SBP; and (q4) whether in patients with AF, treatment effects vary by classes of antihypertensives. Intention-to-treat analysis was adopted using the data provided by each trial, after internal quality checks had been carried out to ensure that data were accurate and transferred without error. Our method for investigating these questions was a one-stage approach that uses IPD from all trials simultaneously and applying a single statistical model. The one-stage approach has more power and flexibility than a two-stage approach to test for treatment-covariate interactions even when few studies are available [15,16]. We used fixed-effect one-stage IPD meta-analysis models for time-to-event data by applying Cox proportional hazard

models stratified by trial [17]. The average SBP reduction between arms among all included trials was 3.7 mm Hg (due to inclusion of "head-to-head" comparisons trials) (S1 Table). Thus, we adjusted the estimates for each subgroup (with and without AF) for a 5-mm Hg reduction in SBP. Furthermore, to assess whether the associations between the intensity of BP reduction and cardiovascular outcomes is similar in those with and without AF at baseline (q2), we used analytical and graphical representations of the full meta-regression model with additional terms for AF status and interactions between treatment, difference in SBP, and AF status. This model describes the effects on outcomes for each level of intensity of SBP lowering and for each of the subgroups with and without AF at baseline. Finally, to assess in patients with AF whether treatment effects vary by baseline SBP and by classes of BP-lowering drugs (q3 and q4), we used models only for AF patients with additional terms for these potential moderators and interactions between treatment, difference in SBP, and moderators (S3 and S4 Tables). Further details on our statistical modelling approach, subgroup analyses, and sensitivity analyses are provided in S1 Methods. Statistical analyses were performed using R version 3.6.1. This study is reported as per the Preferred Reporting Items for Systematic Reviews and Meta-Analyses (PRISMA) guideline (S1 PRISMA Checklist).

## Results

From the full BPLTTC database, 28 trials (145,653 participants) were excluded because AF status at baseline was uncertain or unavailable. Twenty-two trials were eligible and provided data for the IPD meta-analyses (S5 and S6 Tables and Fig 1). Seven of these trials had previously published data about AF status at baseline. The 22 trials included 188,570 individuals, of whom 13,266 (7%) had AF at baseline [12,18–33]. Seven trials explicitly excluded participants with AF at baseline (N = 13,170, 7% of the participants without AF at baseline) [21,34–37], and one trial included only those with prevalent AF (N = 9,016, 67% of the participants with AF at baseline) [12]. The remaining 14 trials included a mixed population of participants with and without AF at baseline (4,249 with AF and 153,198 without AF). All trials contributed data for all the outcomes of interest, with the exception of 2 trials that did not report the HF outcome [19,23] and one trial that did not report cardiovascular death [35]. Risk of bias assessment showed that 20 trials had low risk of bias and 2 trials had moderate risk (S2 Table). There was no evidence of acquisition bias based on funnel plot and Egger's regression test (S1 Fig).

Patients with AF were older than those without AF (mean age 70 versus 65 years, respectively) (Table 1). A lower baseline SBP and diastolic BP (DBP) was evident in patients with AF, who were more commonly prescribed diuretics, angiotensin-converting enzyme inhibitors, beta-blockers, and alpha-blockers: 143/84 mm Hg (SD 21/12 mm Hg) versus 155/88 mm Hg (SD 21/13 mm Hg) in patients without AF, respectively. Cerebrovascular disease was more common in patients with AF, while ischaemic heart disease, diabetes mellitus, and chronic kidney disease were more common in patients without AF. The prevalence of smoking was higher in patients with AF than in those without AF (9.3% versus 24.3%, respectively).

In placebo-controlled trials (8 trials), the difference in SBP reduction between arms was 7.2 (SD 3.9) mm Hg; in drug–drug comparisons (12 trials), it was 2.3 (SD 0.9) mm Hg; and in more versus less intensive BP lowering (2 trials), it was 10.9 (SD 3.0) mm Hg. Overall, the mean difference in SBP reduction between intervention and control arms was 3.7 (SD 3.2) mm Hg, and that was similar in patients with and without AF (3.3 (SD 2.0) mm Hg versus 3.7 (SD 3.3) mm Hg for patients with and without AF, respectively).

Meta-regression showed that there was a linear association between the degree of SBP lowering and the reduction in the hazard ratio (HR) for major cardiovascular events both in patients with and without AF at baseline (Fig 2).

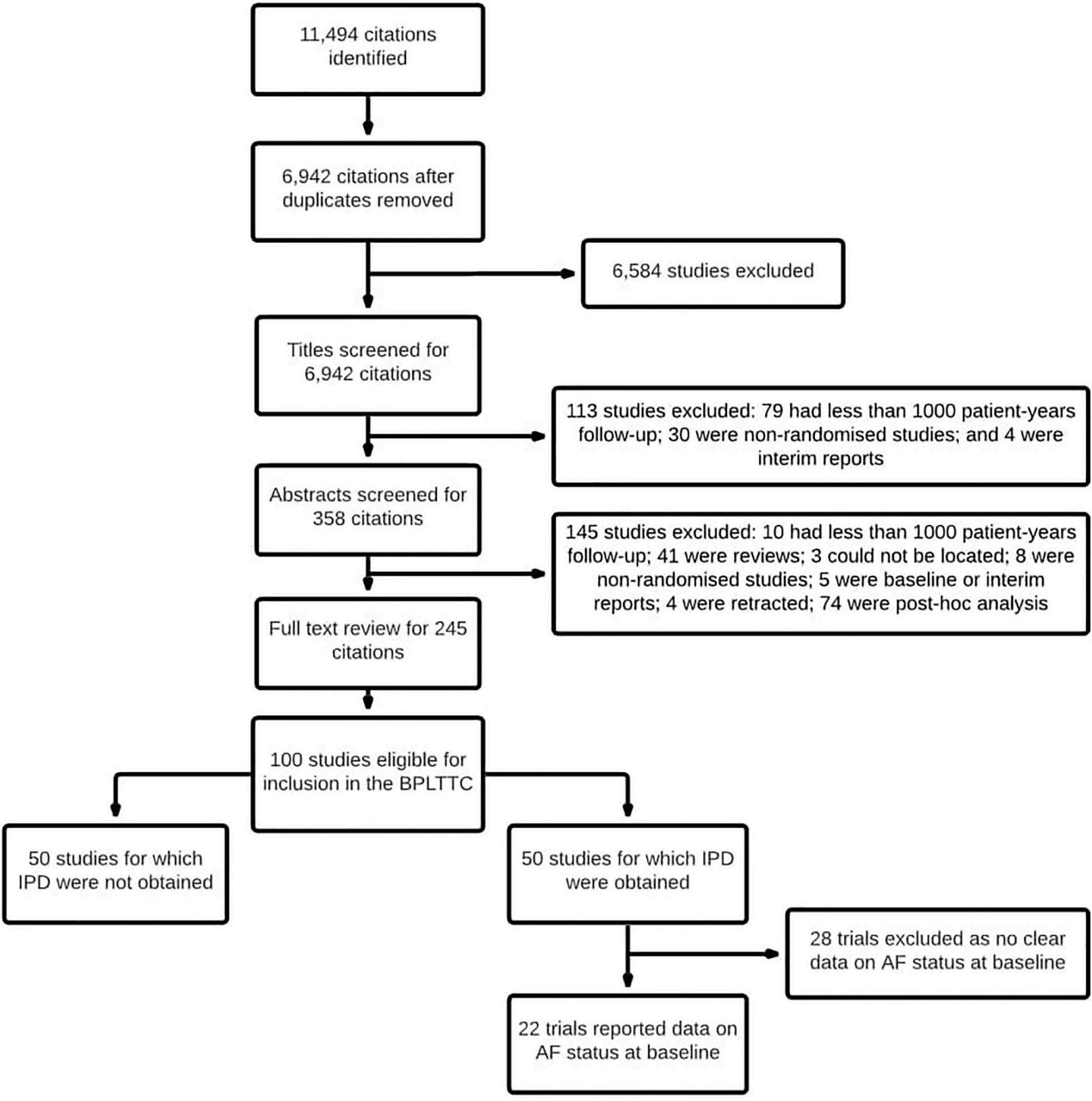

**Fig 1. PRISMA diagram for included trials.** AF, atrial fibrillation; BPLTTC, Blood Pressure Lowering Treatment Trialists' Collaboration; IPD, individual participant data; PRISMA, Preferred Reporting Items for Systematic Reviews and Meta-Analyses.

Over a median follow up of 4.5 years (interquartile range 3.8 to 5.3), 3,674 (27.8%) and 21,380 (12.2%) patients with and without AF, respectively, developed a major cardiovascular event. This translates into a rate of major cardiovascular events of 73 and 28 per 1,000 patient-years for patients with and without AF, respectively (Fig 3). Each 5 mm Hg SBP lowering reduced the risk of major cardiovascular events by about 10% in patients with and without AF at baseline (HR 0.91, 95% confidence interval (CI) [0.83 to 1.00] versus HR 0.91, 95% CI [0.88 to 0.93] for patients with and without AF at baseline, respectively) (Figs 3 and 4). Furthermore, there was no evidence that the risk reduction for any of the primary and secondary outcomes

**Table 1. Baseline characteristics of participants by atrial fibrillation status at baseline.**

|  | Atrial fibrillation (*N* = 13,266) | No atrial fibrillation (*N* = 175,304) | Total (*N* = 188,570) |
|---|---|---|---|
| Age (years) | 70.19 (9.12) | 65.36 (9.05) | 65.70 (9.14) |
| Sex (Female) | 5,052 (38.1) | 73,182 (41.7) | 78,235 (41.5) |
| Ischaemic heart disease | 3,923 (29.6) | 56,759 (32.4) | 60,682 (32.2) |
| Cerebrovascular disease | 2,395 (18.5) | 21,788 (15.5) | 24,183 (15.7) |
| Diabetes mellitus | 3,569 (26.9) | 54,209 (32.2) | 57,778 (31.8) |
| Chronic kidney disease | 163 (20.3) | 15,600 (25.4) | 15,763 (25.3) |
| Heart failure | 2,882 (31.9) | 0 (0) | 2,882 (31.9) |
| Smoking (current) | 1,224 (9.3) | 38,283 (24.3) | 39,507 (23.1) |
| Body mass index (kg/m$^2$) | 28.78 (5.60) | 28.12 (9.67) | 28.16 (9.44) |
| Total cholesterol (mmol/L) | 5.3 (1.2) | 5.6 (1.2) | 5.6 (1.2) |
| Systolic blood pressure (mm Hg) | 142.8 (20.9) | 154.6 (21.6) | 153.82 (21.72) |
| Diastolic blood pressure (mm Hg) | 83.6 (11.7) | 87.8 (12.6) | 87.49 (12.58) |
| Pharmacological treatment |  |  |  |
| Diuretic | 6,082 (50.8) | 19,196 (23.8) | 25,278 (27.3) |
| Alpha-blocker | 1,134 (10.7) | 2,813 (4.4) | 3,947 (5.2) |
| Beta-blocker | 6,133 (51.3) | 29,013 (36.0) | 35,146 (38.0) |
| Angiotensin-converting enzyme inhibitor | 6,846 (59.6) | 32,290 (44.0) | 39,136 (46.1) |
| Angiotensin receptor blocker | 568 (5.4) | 6,420 (15.0) | 6,988 (13.1) |
| Calcium channel blocker | 3,557 (29.7) | 29,566 (36.7) | 33,123 (35.8) |
| Anticoagulant | 4,418 (37.8) | 1,823 (3.1) | 6,241 (8.9) |
| Antiplatelet | 6,443 (56.1) | 35,539 (49.3) | 41,982 (50.2) |
| Lipid-lowering drug | 3,742 (32.8) | 31,100 (42.6) | 34,842 (41.3) |

All categorical variables are summarised as N (% yes); all continuous variables as mean (standard deviation).

achieved by BP-lowering treatment was different between patients with and without AF (Fig 4). Adjustment for the average reduction in SBP of 3.7 mm Hg was consistent with our main adjustment to 5 mm Hg SBP reduction and showed no difference between patients with and without AF (S2 Fig).

Subgroup analysis in patients with AF showed no evidence that the relative risk reduction in major cardiovascular events varied according to baseline SBP (test for linear trend *P* = 0.992). There was also no difference in treatment effects between patients with baseline SBP below and above 140 mmHg (*P* = 0.792) (Fig 5).

Six trials were included in the comparison of renin-angiotensin-aldosterone system (RAAS)-inhibitors versus placebo or standard treatment (i.e., beta-blocker and/or diuretic), including 56,649 participants. Four trials were included in the comparison of calcium channel blockers (CCB)-based regimens versus placebo or standard treatment (i.e., beta-blocker and/or diuretic), including 44,288 participants. There was no evidence of a difference in the effects of treatment regimens based on RAAS inhibitors or CCB between patients with and without AF at baseline (*P* = 0.245 and *P* = 0.909 for RAAS-based and CCB-based regimens, respectively) (Fig 6). However, the CIs were wide due to the relatively small number of AF participants.

Sensitivity analysis using only trials that contributed to both subgroups, that is the 14 trials that included both participants with and without AF at baseline and thus allowed estimation of the within-trial interaction between treatment and AF at baseline, showed broadly similar

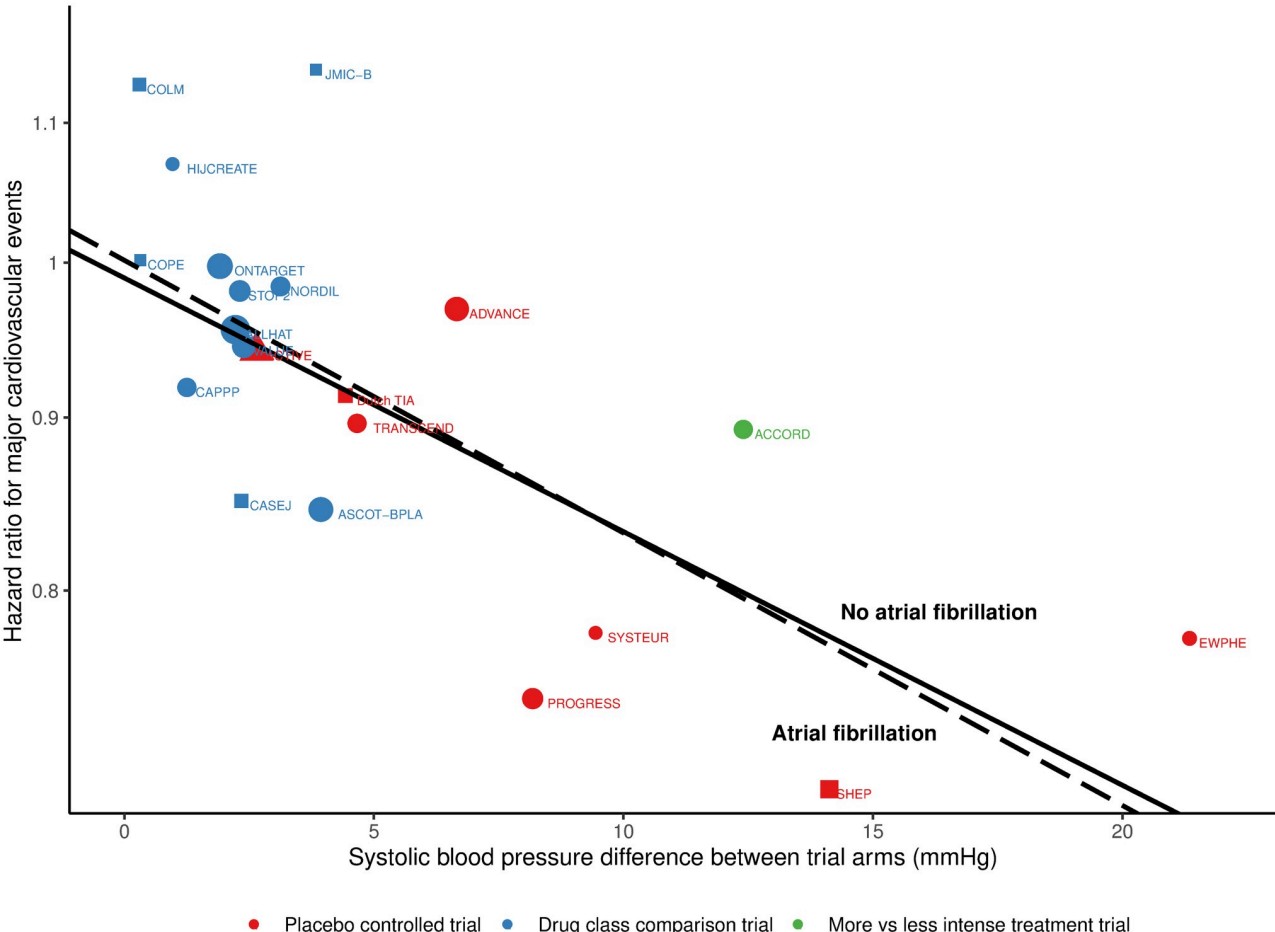

**Fig 2. Hazard ratio of major cardiovascular events related to the 1-year difference blood pressure reduction aggregated at trial level.** Risk of major cardiovascular events, for patients with (dashed line) and without (solid line) atrial fibrillation at baseline, regressed against the systolic blood pressure difference between trial arms, plotted on the log scale. Shapes represent the hazard ratio for each trial with the size inversely proportional to the respective standard error. Trials are coded by shape according to the type of patients: atrial fibrillation only (triangle), no atrial fibrillation only (square), and mix of both (circles). Trials are also coded by colour according to type of intervention: placebo-controlled trials (red), drug class comparison trials (blue), and more versus less intense treatment trials (green). Systolic blood pressure difference between trial arms in mm Hg.

results to those from the main analyses. However, the smaller sample size meant that the CIs were wider, particularly in patients with AF at baseline (S7 Table). Sensitivity analysis using a two-stage approach yielded similar estimates to the one-stage approach overall and for subgroup analysis according to AF status at baseline (S8 Table). Several additional sensitivity analyses were requested by the reviewers which were conducted and broadly supported our main findings: We reran the models using an unadjusted approach for SBP and found no material change in our results (S9 Table). The results of fixed and random effects two-stage models were consistent and further supported the robustness of the main model (S8 Table). To rule out the effects of treatment in ACTIVE-I trial were driven by inclusion of patients with HF, we conducted a sensitivity analysis in which we excluded the patients with a diagnosis of HF at baseline in ACTIVE-I trial. As shown in S10 Table, results were consistent with our main analysis. Additional adjustment for baseline SBP, cardiovascular disease status, and diabetes at baseline had no impact on our main findings (S11 Table). Finally, no material change was also seen after excluding the trials with moderate risk of bias (S12 Table).

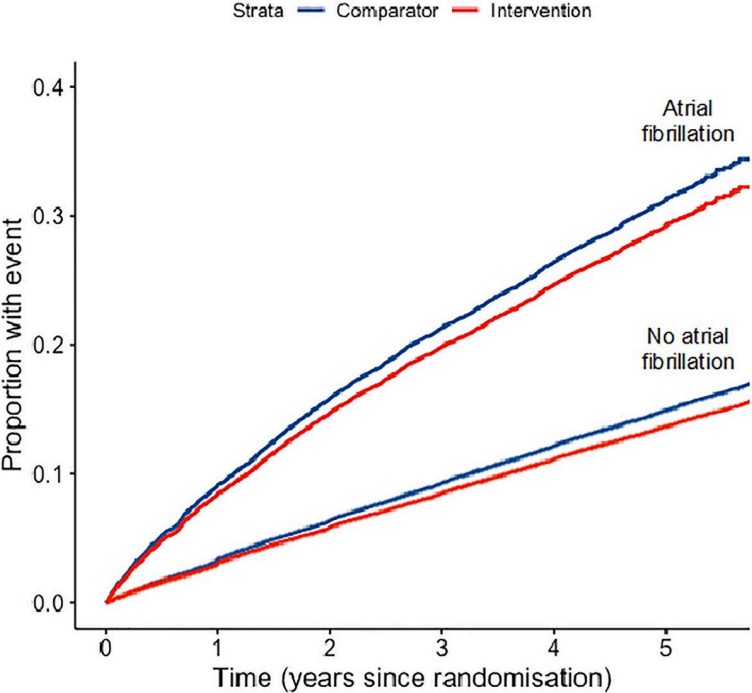

**Fig 3. Cumulative event rates for the primary outcome (major cardiovascular events) by treatment arm, stratified by presence of atrial fibrillation at baseline.** Shown are estimates of the proportions of patients with major cardiovascular events (primary composite endpoint) according to treatment arm (intervention versus comparator as defined in treatment comparisons in the methods) for patients with atrial fibrillation (top lines) and without atrial fibrillation at baseline (bottom lines). These curves were created for the overall population included in this study without accounting for stratification by trial.

## Discussion

This study showed that BP-lowering treatment affords a similar relative risk reduction in major cardiovascular events in patients with and without AF, with no evidence that treatment effects differed between those subgroups for any of the primary and secondary outcomes. Overall, each 5-mm Hg reduction in SBP resulted in an approximately 10% lower risk of major cardiovascular events both in patients with and without AF at baseline. Furthermore, there was no evidence that in patients with AF the relative risk reduction varied according to baseline SBP [38].

Although absolute risks are better estimated from population-based observational studies, the almost 3-fold higher event rate that we observed in patients with AF at baseline compared with those without AF reflects their higher cardiovascular risk. This is in keeping with previous observational studies which reported that AF was associated with a 2- to 5-fold higher risk of major cardiovascular events in comparison with patients without AF [4,39,40]. Therefore, the same relative risk reduction afforded by BP-lowering treatment would most likely achieve a greater absolute risk reduction in patients with AF than in patients without AF at baseline.

It is thus a paradox that much of the focus of AF-related research has been on anticoagulation for stroke prevention, and strategies for rate-control or restoration of sinus rhythm, when the relative and absolute risk for cardiovascular events like HF and ischaemic heart disease is greater than that of stroke in patients with AF [40]. In addition, even with optimal anticoagulation and rate or rhythm control, the risk of stroke in patients with AF remains high (about 1.5% per year) [41] and it seems to result from associated risk factors rather than treatment

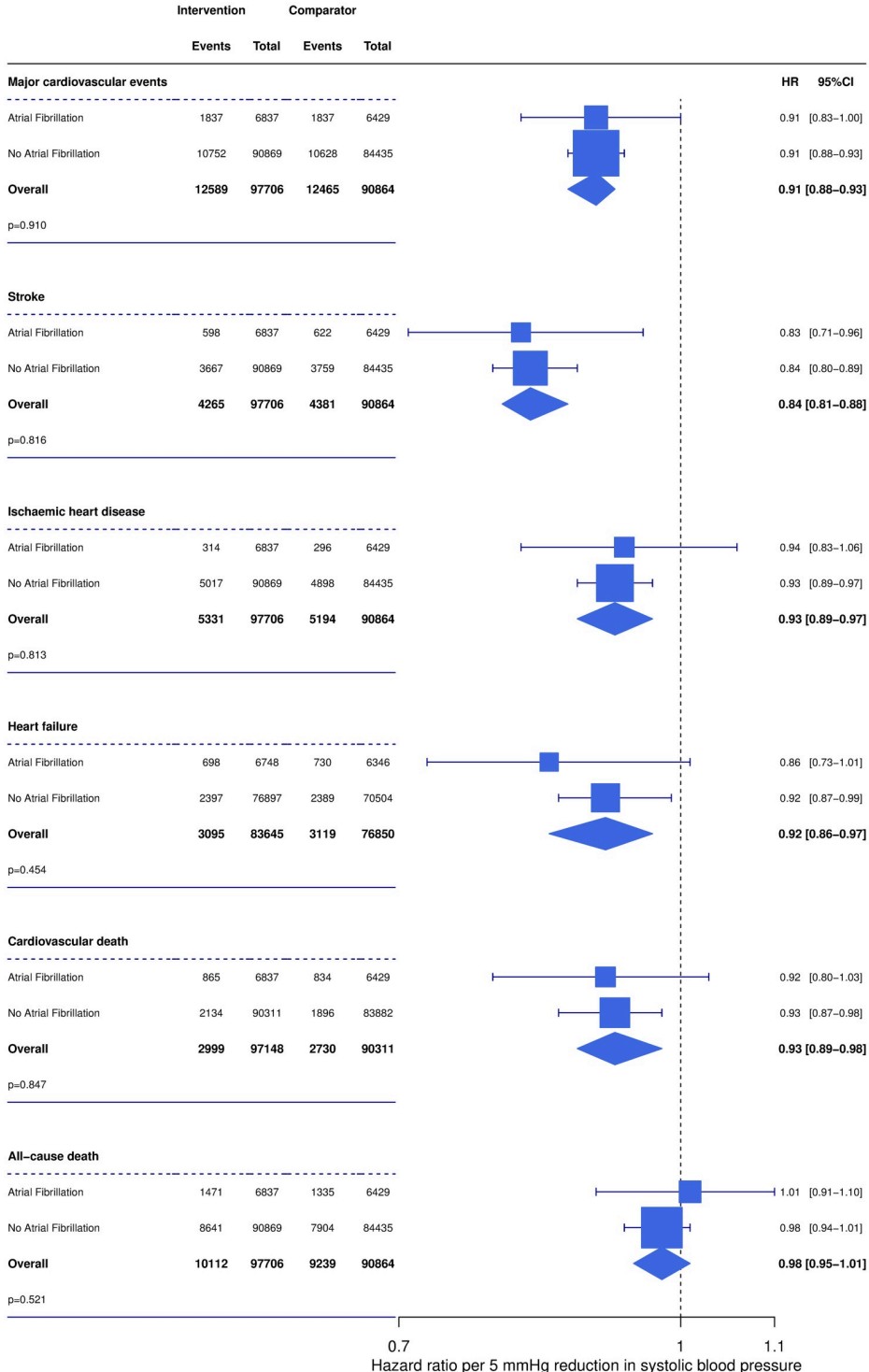

**Fig 4. Effect of blood pressure lowering treatment on primary and secondary outcomes, stratified by presence of atrial fibrillation at baseline.** Forest plot displays the HRs and 95% CIs for each outcome adjusted for a 5-mm Hg systolic blood pressure reduction. Further details on adjustment provided in the Methods. *P* values for test of difference between subgroups. CI, confidence interval; HR, hazard ratio.

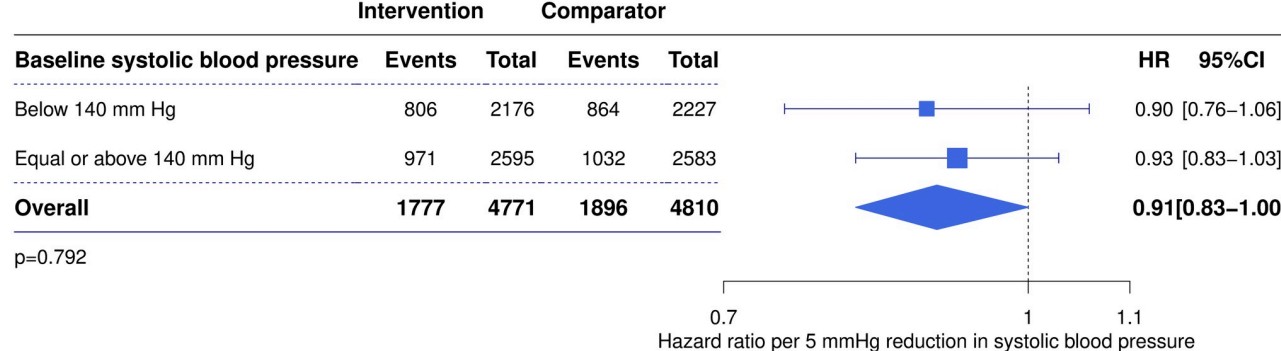

**Fig 5. Effect of blood pressure-lowering treatment on major cardiovascular events stratified by baseline systolic blood pressure in patients with atrial fibrillation.** Forest plot displays the HRs and 95% CIs for major cardiovascular events for a 5-mm Hg systolic blood pressure reduction in patients with atrial fibrillation with baseline systolic blood pressure below or above 140 mm Hg. *P* value for test of difference between subgroups. CI, confidence interval; HR, hazard ratio.

failure [42,43]. Therefore, management of associated cardiovascular risk factors, among which high BP with an estimated prevalence of 70% is the most common, seems a priority to improve cardiovascular outcomes and survival in the high-risk group of patients with AF [44]. In this context, our study provides compelling evidence that pharmacological BP-lowering treatment is an effective strategy to prevent cardiovascular events overall as well as to address the residual risk of stroke.

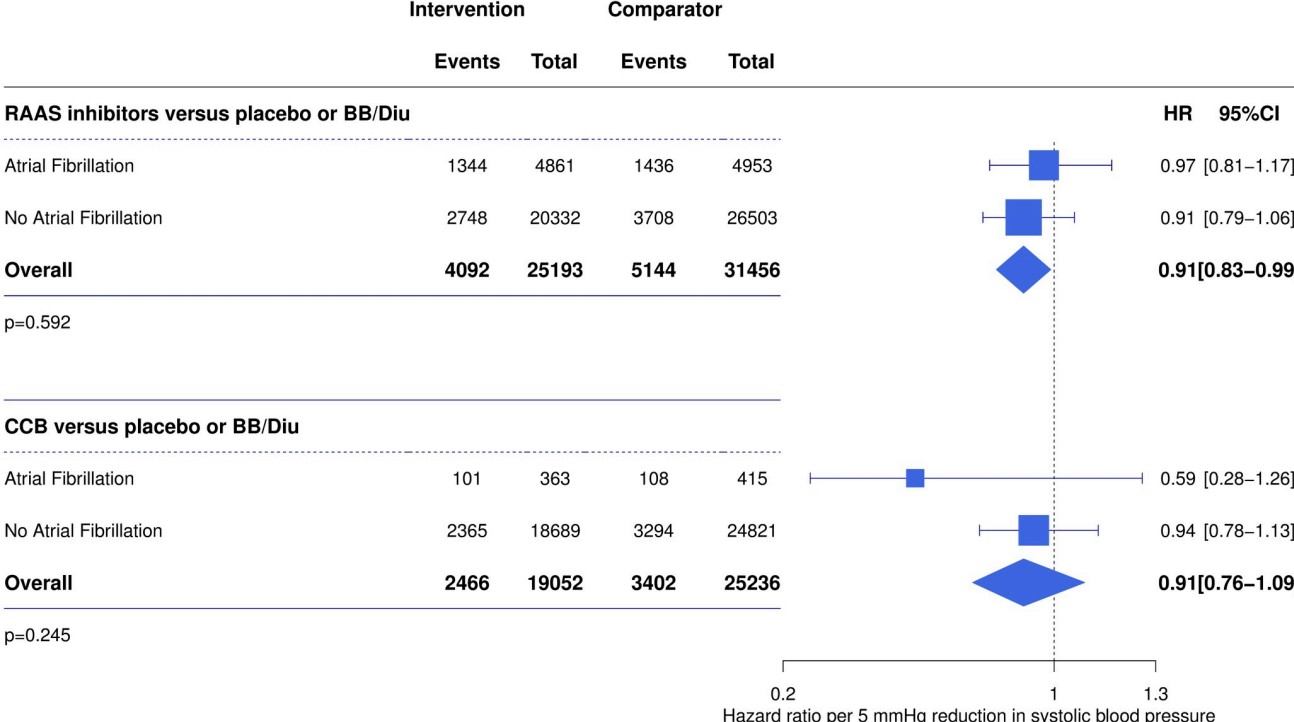

**Fig 6. Effect of blood pressure-lowering treatment on major cardiovascular events stratified by drug class.** Forest plot displays the HRs and 95% CIs for major cardiovascular events for a 5-mm Hg systolic blood pressure reduction for RAAS inhibitors-based and CCB-based regimens in comparison with placebo or BB with or without Diu. *P* values for test of difference between subgroups. BB, beta-blocker; CCB, calcium channel blocker; CI, confidence interval; Diu, diuretic; HR, hazard ratio; RAAS, renin-angiotensin-aldosterone system.

Although the most recent AF guidelines issued by the European Society of Cardiology state that "good blood pressure control should form an integral part of the management of AF patients," randomised evidence has been lacking to support those recommendations [7]. This uncertainty underpins the cautious AF guidelines of the American College of Cardiology, which despite mentioning that "appropriate control of risk factors like hypertension substantially reduces stroke risk," make no specific recommendations about BP management in patient with AF [9]. Thus far, evidence on potential importance of BP reduction in patients with AF comes from observational studies, wherein the ideal SBP target for the prevention of cardiovascular events was found to be in the range of 120 to 129 mmHg [4]. Our study fills this gap in randomised evidence. We found that patients with AF included in trials had a relatively low baseline SBP, with almost half of them having a SBP of less than 140 mmHg prior to randomisation, commonly not recommended for treatment [38].

Hypertension guidelines recommend that drugs with shared rate- and BP-lowering properties (e.g., non-dihydropyridine CCBs and beta-blockers) should be preferred in patients with AF and high BP [38,45]. However, those recommendations are based on the indication of those drugs for rate control as thus far whether pharmacological BP lowering decreases cardiovascular risk in patients with AF has not been demonstrated. In addition, the lack of evidence for class-specific effects together with the linear association between the HR for major cardiovascular events and the intensity of SBP reduction showed in this study suggest that the intensity of BP lowering is more important than the specific drugs used to achieve it when it comes to prevention cardiovascular events. Therefore, until further research clarifies whether any drug class can achieve superior risk reduction for equivalent BP reduction, BP lowering irrespective of the drug class should be viewed as an effective strategy to decrease the high cardiovascular risk of patients with AF.

The main strength of this IPD meta-analysis is the analysis of the effects of BP-lowering treatment in a large number of patients with baseline AF included into RCTs, their comparison with patients without AF, the long follow-up time, and the more than 20,000 major cardiovascular events. The robustness of the main conclusions to sensitivity analyses and the consistency of the estimates provided by different methods further support the conclusions drawn. However, some limitations deserve to be acknowledged. First, there is a possibility of selection bias, as IPD was not obtainable for all the trials eligible for inclusion in the BPLTTC. While it remains uncertain whether those trials had collected data on AF status at baseline, it seems unlikely that lack of contribution to the BPLTTC has biased our estimates. This is supported by the consistency of the effects of BP reduction on cardiovascular outcomes in our set of studies with the estimates in a more comprehensive, albeit tabular meta-analysis (roughly 10% relative risk reduction per 5 mm Hg SBP reduction) [10]. Second, the number of participants with AF was modest considering the total number of participants in the BPLTTC dataset because only a fraction of the trials reported AF at baseline and the relatively low rate of AF in the trial populations. On the other hand, this means that even if a degree of misclassification is present due to omitted disclosure or the paroxysmal nature of the arrhythmia, a material impact on treatment effect estimates would be unlikely. Third, although it would have been interesting to compare the effects of other drug classes, particularly in head-to-head comparisons, only a small fraction of the trials included in the BPLTTC reported AF at baseline and were thus eligible for this study. Fourth, although concerns have been raised about the variability of BP measurement in patients with AF, which could have biased our estimates, there was no evidence of this in our population [46]. Finally, much of the weight of the analysis in the current report was driven by the ACTIVE-I trial, which included about 30% of patients with HF. However, excluding the HF patients from the analysis did not have a material impact on

our findings of no interaction between AF status at baseline and the effect of BP lowering treatment.

In conclusion, this study demonstrated that BP-lowering treatment reduces the risk of major cardiovascular events in patients with AF to a similar extent to that of patients without AF, with no evidence that treatment effects varied according to baseline SBP or drug class. Owing to their higher absolute cardiovascular risk, treatment in patients with AF is likely to result in greater absolute risk reduction than in patients without AF. Clinical guidelines should be updated to recommend pharmacological BP lowering for prevention of cardiovascular events in patients with AF.

## Supporting information

**S1 Fig. Funnel plot for assessment of publication (acquisition) bias on the effect of blood pressure reduction and risk of major cardiovascular event.**
(DOCX)

**S2 Fig. Sensitivity analysis for the effect of blood pressure-lowering treatment on primary and secondary outcomes, stratified by presence of atrial fibrillation at baseline and adjusted for a 3.7-mm Hg systolic blood pressure reduction.**
(DOCX)

**S1 Table. Difference in systolic blood pressure reduction between arms for each trial.**
(DOCX)

**S2 Table. Assessment of risk of bias.**
(DOCX)

**S3 Table. Number of trials available for drug class comparisons.**
(DOCX)

**S4 Table. Treatment comparisons for subgroup analyses by drug class.**
(DOCX)

**S5 Table. Summary of included trials.**
(DOCX)

**S6 Table. Baseline characteristics of the participants included in atrial fibrillation meta-analyses stratified by trial.**
(DOCX)

**S7 Table. Sensitivity analyses including only trials that included patients with and without atrial fibrillation at baseline.**
(DOCX)

**S8 Table. Fixed and random effects two-stage meta-analyses.**
(DOCX)

**S9 Table. Unadjusted effect of blood pressure-lowering treatment on primary and secondary outcomes, stratified by the presence of atrial fibrillation at baseline.**
(DOCX)

**S10 Table. Sensitivity analysis for the effect of blood pressure-lowering treatment on primary and secondary outcomes, stratified by the presence of atrial fibrillation at baseline, excluding the patients with the diagnosis of heart failure at baseline in ACTIVE-I trial.**
(DOCX)

**S11 Table. Sensitivity analysis for the effect of blood pressure-lowering treatment on primary and secondary outcomes, stratified by the presence of atrial fibrillation at baseline, after adjustment for baseline systolic blood pressure, cardiovascular disease status, and diabetes status at baseline.**
(DOCX)

**S12 Table. Sensitivity analysis for the effect of blood pressure-lowering treatment on primary and secondary outcomes, stratified by the presence of atrial fibrillation at baseline, excluding the trials with moderate risk of bias.**
(DOCX)

**S1 Methods.**
(DOCX)

**S1 PRISMA Checklist.**
(DOCX)

**S1 Protocol.**
(DOCX)

## Acknowledgments

**Writing group**: Ana-Catarina Pinho-Gomes (King's College London, London, United Kingdom), Luis Azevedo (Centre for Health Technology and Services Research, University of Porto, Porto, Portugal), Emma Copland (Deep Medicine, Nuffield Department of Women's and Reproductive Health, University of Oxford, Oxford, United Kingdom), Dexter Canoy (Deep Medicine, Nuffield Department of Women's and Reproductive Health, University of Oxford, Oxford, United Kingdom), Milad Nazarzadeh (Deep Medicine, Nuffield Department of Women's and Reproductive Health, University of Oxford, Oxford, United Kingdom), Rema Ramakrishnan (Deep Medicine, Nuffield Department of Women's and Reproductive Health, University of Oxford, Oxford, United Kingdom), Eivind Berge (Department of Cardiology, Oslo University Hospital, Oslo, Norway),†Johan Sundström (Department of Medical Sciences, Uppsala University, Sweden),[7] Dipak Kotecha (Institute of Cardiovascular Sciences, University of Birmingham, Birmingham, United Kingdom), Mark Woodward (The George Institute for Global Health, University of New South Wales, Sydney, Australia), Koon Teo (Population Health Research Institute, McMaster University, Hamilton, Ontario, Canada), Barry R Davis (The University of Texas School of Public Health, Houston, Texas, United States),[13] John Chalmers (The George Institute for Global Health, University of New South Wales, Sydney, Australia),[9] Carl J. Pepine (Department of Medicine, University of Florida, Gainesville, Florida, United States),[14] Kazem Rahimi (Deep Medicine, Nuffield Department of Women's and Reproductive Health, University of Oxford, Oxford, United Kingdom))[3,4].

† Deceased.

**Core analytic group**: Zeinab Bidel, Milad Nazarzadeh, Emma Copland and Dexter Canoy (Deep Medicine, Nuffield Department of Women's and Reproductive Health, University of Oxford, Oxford, United Kingdom).

**Steering Committee**: Kazem Rahimi (Chair) (Deep Medicine, University of Oxford, Oxford, United Kingdom), Koon Teo (Population Health Research Institute, McMaster University, Hamilton, Ontario, Canada) Barry R Davis (The University of Texas School of Public Health, Houston, Texas, USA), John Chalmers (The George Institute for Global Health, University of New South Wales, Sydney, Australia), Carl J. Pepine (Department of Medicine, University of Florida, Gainesville, Florida, USA).

**Collaborating Trialists**: A Adler (UKPDS [UK Prospective Diabetes Study]), L Agodoa (AASK [African-American Study of Kidney Disease and Hypertension]), A Algra (Dutch TIA Study [Dutch Transient Ischemic Attack Study]), F W Asselbergs (PREVEND-IT [Prevention of Renal and Vascular End- stage Disease Intervention Trial]), N Beckett (HYVET [Hypertension in the Very Elderly Trial]), E Berge (deceased) (VALUE trial [Valsartan Antihypertensive Long-term Use Evaluation trial]), H Black (CONVINCE [Controlled Onset Verapamil Investigation of Cardiovascular End Points]), F.P.J. Brouwers (PREVEND-IT), M Brown (INSIGHT [International Nifedipine GITS Study: Intervention as a Goal in Hypertension]), C J Bulpitt (HYVET), B Byington (PREVENT [Prospective Randomized Evaluation of the Vascular Effects of Norvasc Trial]), J Chalmers (ADVANCE [Action in Diabetes and Vascular Disease: Preterax and Diamicron MR Controlled Evaluation]), WC Cushman ((ACCORD [Action to Control Cardiovascular Risk in Diabetes], ALLHAT, SPRINT [Systolic Blood Pressure Intervention Trial]), J Cutler (ALLHAT [Antihypertensive and Lipid-Lowering Treatment to Prevent Heart Attack Trial]), B R Davis (ALLHAT), R B Devereaux (LIFE [Losartan Intervention For Endpoint reduction in hypertension]), D Dwyer (IDNT [Irbesartan Diabetic Nephropathy Trial]), R Estacio (ABCD [Appropriate Blood Pressure Control in Diabetes]), R Fagard (SYST-EUR [SYSTolic Hypertension in EURope]), K Fox (EUROPA [European trial on Reduction Of cardiac events with Perindopril among patients with stable coronary Artery disease]), T Fukui (CASE-J [Candesartan Antihypertensive Survival Evaluation in Japan]), A K Gupta (ASCOT [Anglo-Scandinavian Cardiac Outcomes Trial]), R R Holman (UKPDS [UK Prospective Diabetes Study]), Y Imai (HOMED-BP [Hypertension Objective Treatment Based on Measurement by Electrical Devices of Blood Pressure]), M Ishii (JMIC-B [Japan Multicenter Investigation for Cardiovascular Diseases-B]), S Julius (VALUE), Y Kanno (E-COST [Efficacy of Candesartan on Outcome in Saitama Trial]), S E Kjeldsen (VALUE, LIFE), J Kostis (SHEP [Systolic Hypertension in the Elderly Program]) K Kuramoto (NICS-EH [National Intervention Cooperative Study in Elderly Hypertensives]), J Lanke (STOP2 [Swedish Trial in Old Patients with Hypertension-2], NORDIL [Nordic Diltiazem]), E Lewis (IDNT), J B Lewis (IDNT) M Lievre (DIABHYCAR [Non-insulin-dependent diabetes, hypertension, microalbuminuria or proteinuria, cardiovascular events, and ramipril study]), L H Lindholm (CAPPP [Captopril Prevention Project], STOP2, NORDIL), S Lueders (MOSES [The Morbidity and Mortality After Stroke, Eprosartan Compared With Nitrendipine for Secondary Prevention]), S MacMahon (ADVANCE), G Mancia (INSIGHT), M Matsuzaki (COPE [The Combination Therapy of Hypertension to Prevent Cardiovascular Events]), M H Mehlum (VALUE), S Nissen (CAMELOT [Comparison of Amlodipine vs Enalapril to Limit Occurrences of Thrombosis]), H Ogawa (HIJ-CREATE [Heart Institute of Japan Candesartan Randomized Trial for Evaluation in Coronary Heart Disease]), T Ogihara (CASE-J), T Ohkubo (HOMED-BP), C Palmer (INSIGHT), A Patel (ADVANCE), C J Pepine (INVEST [International Verapamil SR-Trandolapril Study]), M Pfeffer (PEACE [Prevention of Events With Angiotensin- Converting Enzyme Inhibition]), B Pitt (PREVENT), N R Poulter (ASCOT [Anglo-Scandinavian Cardiac Outcomes Trial]), H Rakugi (VALISH [Valsartan in Elderly Isolated Systolic Hypertension Study], CASE-J), G Reboldi (Cardio-Sis [CARDIOvascolari del Controllo della Pressione Arteriosa SIStolica]), C Reid (ANBP2 [The Second Australian National Blood Pressure Study]), G Remuzzi (BENEDICT [BErgamo NEphrologic DIabetes Complications Trial]), P Ruggenenti (BENEDICT), T Saruta (CASE-J), J Schrader (MOSES), R Schrier (deceased) (ABCD), P Sever (ASCOT), P Sleight (deceased) (CONVINCE, HOPE [Heart Outcomes Prevention Evaluation], TRANSCEND [Telmisartan Randomised AssessmeNt Study in ACE iNtolerant subjects with cardiovascular Disease], ONTARGET [Ongoing Telmisartan Alone and in Combination with Ramipril Global Endpoint Trial]), J A Staessen (SYST-EUR [Systolic Hypertension in Europe]), H Suzuki (ECOST), L Thijs (Syst-Eur), K Ueshima (VALISH,

CASE-J), S Umemoto (COPE), W H van Gilst (PREVEND-IT), P Verdecchia (Cardio-Sis [CARDIOvascolari del Controllo della Pressione Arteriosa SIStolica]), K Wachtell (LIFE), P Whelton (SPRINT), L Wing (ANBP2 [The Second Australian National Blood Pressure Study]), M Woodward (ADVANCE, PROGRESS), Y Yui (JMIC-B), S Yusuf (HOPE, ONTARGET, TRANSCEND), A Zanchetti (deceased) (VHAS [Verapamil in Hypertension and Atherosclerosis Study], ELSA [European Lacidipine Study on Atherosclerosis]), Z Y Zhang (Syst-Eur).

**Other members**: C Anderson, C Baigent, BM Brenner, R Collins, D de Zeeuw, J Lubsen, E Malacco, B Neal, V Perkovic, A Rodgers, P Rothwell, G Salimi-Khorshidi, J Sundström, F Turnbull, G Viberti, J Wang.

## Author Contributions

**Conceptualization:** Ana-Catarina Pinho-Gomes, Luis Azevedo, Eivind Berge, Johan Sundström, Dipak Kotecha, Mark Woodward, Kazem Rahimi.

**Data curation:** Ana-Catarina Pinho-Gomes, Emma Copland, Dexter Canoy.

**Formal analysis:** Ana-Catarina Pinho-Gomes, Luis Azevedo, Mark Woodward.

**Funding acquisition:** Ana-Catarina Pinho-Gomes, Kazem Rahimi.

**Investigation:** Ana-Catarina Pinho-Gomes, Luis Azevedo, Dexter Canoy.

**Methodology:** Ana-Catarina Pinho-Gomes, Luis Azevedo, Johan Sundström, Mark Woodward.

**Project administration:** Ana-Catarina Pinho-Gomes, Kazem Rahimi.

**Resources:** Ana-Catarina Pinho-Gomes.

**Software:** Ana-Catarina Pinho-Gomes.

**Supervision:** Luis Azevedo, Mark Woodward, Kazem Rahimi.

**Validation:** Luis Azevedo.

**Visualization:** Ana-Catarina Pinho-Gomes, Luis Azevedo, Mark Woodward, Kazem Rahimi.

**Writing – original draft:** Ana-Catarina Pinho-Gomes.

**Writing – review & editing:** Ana-Catarina Pinho-Gomes, Luis Azevedo, Dexter Canoy, Milad Nazarzadeh, Rema Ramakrishnan, Eivind Berge, Johan Sundström, Dipak Kotecha, Mark Woodward, Koon Teo, Barry R. Davis, John Chalmers, Carl J. Pepine, Kazem Rahimi.

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
