## [Editor Report · Decision Letter 0]

1 May 2020

Dear Dr Rahimi, 

Thank you for submitting your manuscript entitled "Blood pressure lowering treatment for prevention of cardiovascular events in patients with atrial fibrillation: an individual-participant data meta-analysis" for consideration by PLOS Medicine.

Your manuscript has now been evaluated by the PLOS Medicine editorial staff [as well as by an academic editor with relevant expertise] and I am writing to let you know that we would like to send your submission out for external peer review.

Kind regards,

Adya Misra, PhD,

Senior Editor

PLOS Medicine

---

## [Decision Letter · Decision Letter 1]

15 Jul 2020

Dear Dr. Rahimi,

Thank you very much for submitting your manuscript "Blood pressure lowering treatment for prevention of cardiovascular events in patients with atrial fibrillation: an individual-participant data meta-analysis" (PMEDICINE-D-20-01625R1) for consideration at PLOS Medicine. 

[LINK]

In light of these reviews, I am afraid that we will not be able to accept the manuscript for publication in the journal in its current form, but we would like to consider a revised version that addresses the reviewers' and editors' comments. Obviously we cannot make any decision about publication until we have seen the revised manuscript and your response, and we plan to seek re-review by one or more of the reviewers. 

We expect to receive your revised manuscript by Aug 05 2020 11:59PM. Please email us (plosmedicine@plos.org) if you have any questions or concerns.

We look forward to receiving your revised manuscript. 

Sincerely,

Emma Veitch, PhD

PLOS Medicine

On behalf of Clare Stone, PhD, Acting Chief Editor,

PLOS Medicine

plosmedicine.org

*Although two reviewers are relatively positive about the paper, one reviewer (reviewer 2) raises substantial concerns about the selectivity of inclusion of data in this IPD meta, among other issues - for the paper to be considered potentially publishable in PLOS Medicine, the authors would need to come up with a convincing rebuttal to those points. In addition although the authors' Discussion limitations section mentions some aspects of the analyses, the exact point raised by this reviewer does not seem to be mentioned there. 

*Some minor style changes (to be addressed alongside the reviewers' points) - In the last sentence of the Abstract Methods and Findings section, please include a brief note about any key limitation(s) of the study's methodology.

*At this stage, we ask that you include a short, non-technical Author Summary of your research to make findings accessible to a wide audience that includes both scientists and non-scientists. The Author Summary should immediately follow the Abstract in your revised manuscript. This text is subject to editorial change and should be distinct from the scientific abstract. Please see our author guidelines for more information: https://journals.plos.org/plosmedicine/s/revising-your-manuscript#loc-author-summary

*We would ask that the authors state clearly in the Methods section of the paper whether the analysis presented here corresponds to one laid out in a prospectively-specified protocol or analysis plan? Please state this (either way) early in the Methods section.

Comments from the reviewers:

Reviewer #1: See attachment

Michael Dewey

Reviewer #2: Dear Editors of PLOS Medicine,

Thank you for the opportunity to review this manuscript by Prof. Kazem Rahimi and colleagues. This is an individual-participant data meta-analysis from the Blood Pressure Treatment Trialists Collaboration (BPLTTC), aiming to assess if the effect of blood pressure (BP) lowering treatment differs between people with and without atrial fibrillation (AF), and if treatment effect in people with AF differs by baseline BP level and choice of antihypertensive agent. 

Firstly, the authors should be complemented for the effort to update of the BPLTTC database. This will surely be of great importance to the field for years to come. Further, the manuscript is well written, with clear graphs and tables. 

Unfortunately, I have several major concerns that needs to be addressed. Firstly, this meta-analysis does not build on a systematic review of the literature. This is problematic, because inclusion of certain (not all of the available) trials may provide biased results, not representative of the totality of the available evidence. For example, the five-year-old meta-analysis by Ettehad et al.1, with similar inclusion criteria, included 123 trials, whereas the updated BPLTTC database includes 50 trials, of which 22 are included in this analysis. Without an underlying systematic review, it is not possible to assess the generalizability of the reported findings. When undertaking such a massive effort as updating the BPLTTC dataset, it would be a small thing to perform an updated systematic review, trying to find all available trials, thereby being able to assess the representativeness of the trials providing IPD. 

Secondly, also relating to the fact that this is not a systematic review, the authors present no risk of bias assessment of the included trials. This is a major concern, primarily because it makes it impossible for the reader to appraise the validity of the results from the meta-analysis (a meta-analysis is only as good as the included trials, and this is not reported), but also because it is a deviation from the meta-analysis protocol, published last year.2

Thirdly, the main advantage of IPD in a meta-analysis setting is to better be able to select and group participants, based on individual characteristics. Used correctly, this will minimize ecological bias and confounding by differences in baseline covariates between groups. This paper hasn't used the available IPD as efficient as possible. For example, in the ACTIVE-I trial (I'll come back to that), contributing with 67% of AF participants in this meta-analysis, one third of participants had heart failure (HF) at baseline.3 These participants likely had lower BP and benefited more from treatment compared to non-HF participants; and would have been managed according to HF guidelines (not hypertension guidelines) in clinical practice. Including such participants in an IPD meta-analysis on the effects of blood pressure lowering treatment will I) bias the results towards more positive findings in people with AF and low BP, II) impair the clinical usefulness of the results because they do not necessarily apply to non-HF patients. 

Fourthly, inclusion of trials in which all participants had AF, and trials where none of the included participants had AF, greatly increases the risk of ecological bias. If only trials including both AF and non-AF participants would have been included, AF and non-AF participants would have been judged according to the same eligibility criteria, received the same intervention and control, and the same follow-up procedures, including event adjudication etc. Differences in baseline characteristics would still have been present, but these would have been small, and at least trial-level factors would have been similar. In the current analysis, the majority of participants in the AF group comes from one trial (ACTIVE-I), which was limited to AF participants only. This results in massive differences in age, baseline BP level, and concomitant treatment between groups. Further, it means AF and non-AF participants in the meta-analysis have been judged according to different eligibility criteria (trial level), received different interventions, and different follow-up. 

Fifthly, several decisions and interpretations relating the statistical methods are questionable. 

1. The authors (correctly) state that "Due to different settings and patient populations, the trials included in a meta-analysis are likely to have different baseline hazard functions…" as a motivation for why they stratified their Cox model. In the next sentence they go on stating they chose fixed-effects model for their meta-analysis. This makes no sense, given that the fixed-effects model builds on the assumption of one common treatment effect in all trials. This seems careless rather than "parsimonious", which is the argument used in the supplementary methods section. Cornell et al. discuss how to estimate uncertainty in case of heterogeneity (which was present in the current analysis).4 

2. The authors "standardize" the results of each trial to 5 mmHg BP difference between groups. In the best-case scenario, this exaggerates the estimated treatment effect by about 30% (because the actual difference was 3.7 mmHg). In the worst-case scenario, this causes biased results due to shifts in weight between trials, as shown by Brunström et al. and discussed by Thomopoulos.5, 6

3. At several occasions, the authors use underpowered analyses to motivate their choices of statistical methods, and to draw conclusions about possible interactions. For example, heterogeneity analyses should never be used to decide whether to use fixed- or random-effects model.7 Especially in situations like this, when the sample size of AF patients is relatively small, the test for heterogeneity will fail to detect fairly large differences in treatment effects because of the large standard error in the AF groups. Further, research question 3 and 4, regarding the possible interaction between baseline BP and treatment effect, and choice of agent and treatment effect, cannot be answered by the available data. As is evident from Figure 4 and 5, confidence intervals are wide, e.g. spanning from 0.76 to 1.06 for the <140 mmHg group in the BP analysis, reflecting the low number of participants and events. Indeed "no evidence for an association" is not the same as "evidence for no association"; this is not reflected in the authors' interpretations. Specifically, the conclusion that BP lowering is beneficial in AF patients, even below SBP 140 mmHg, stated in the abstract as well as the full text conclusion, is not supported by the data. 

Minor issues:

- Inclusion criteria are different in abstract and methods; 1000 patient-years per trial or treatment arm? 

- Numbers in fig 4 does not add up. 2982+3566 ≠ 6837; 3091+3615 ≠ 6429 etc. Numbers in BP strata are not compatible with numbers in Fig 3. 

1. Ettehad D, Emdin CA, Kiran A, et al. Blood pressure lowering for prevention of cardiovascular disease and death: a systematic review and meta-analysis. Lancet 2016; 387: 957-67.

2. Rahimi K, Canoy D, Nazarzadeh M, et al. Investigating the stratified efficacy and safety of pharmacological blood pressure-lowering: an overall protocol for individual patient-level data meta-analyses of over 300 000 randomised participants in the new phase of the Blood Pressure Lowering Treatment Trialists' Collaboration (BPLTTC). BMJ Open 2019; 9: e028698.

3. Yusuf S, Healey JS, Pogue J, et al. Irbesartan in Patients with Atrial Fibrillation. New England Journal of Medicine 2011; 364: 928-38.

4. Cornell JE, Mulrow CD, Localio R, et al. Random-Effects Meta-analysis of Inconsistent Effects: A Time for Change. Annals of Internal Medicine 2014; 160: 267-70.

5. Brunström M, Carlberg B. Standardization according to blood pressure lowering in meta-analyses of antihypertensive trials: comparison of three methodological approaches. J Hypertens 2018; 36: 4-15.

6. Thomopoulos C, Michalopoulou H. Outcome standardization to blood pressure reduction in meta-analyses: sailing in uncharted waters. J Hypertens 2018; 36: 31-3.

7. Cornell JE, Mulrow CD, Localio R, et al. Random-effects meta-analysis of inconsistent effects: a time for change. Ann Intern Med. 2014; 160(4):267-270. 

Reviewer #3: this study aimed to compare the effects of blood pressure (BP)-lowering drugs in patients with and without atrial fibrillation (AF) at baseline. The authors conclude that BP-lowering treatment reduces the risk of major cardiovascular events similarly in individuals with and without AF. Pharmacological BP lowering for prevention of cardiovascular events should be recommended in patients with AF even if BP is below the conventional 140-mmHg threshold.

This paper is generally nicely written and analysed

While there has not been a specific RCT in AF, some large observational studies on the optimal BP targets have been published eg from the Yonsei group in S Korea (Joung B and team).

These should be discussed in the Discusion, notwithstanding that observational cohorts are not a substitute for RCTs

Figure 4 is simplistic - BP is not a yes/no parameter - show the impact on outcomes with BP as a continuous variable, for stroke, major CV events, all cause mortality etc

Figure 5 should how lump together 'placebo or BB/Diu'

Ideally show a summary forrest plot of the different drug classes vs placebo, even to illustrate the effect sizes

Forest plot can also show inter-drug class comparisons

[LINK]

---

## [Decision Letter · Decision Letter 2]

26 Oct 2020

Dear Dr. Rahimi,

Thank you very much for submitting your manuscript "Blood pressure lowering treatment for prevention of cardiovascular events in patients with atrial fibrillation: an individual-participant data meta-analysis" (PMEDICINE-D-20-01625R2) for consideration at PLOS Medicine. 

[LINK]

In light of these reviews, I am afraid that we will not be able to accept the manuscript for publication in the journal in its current form, but we would like to consider a revised version that addresses the reviewers' and editors' comments. Obviously we cannot make any decision about publication until we have seen the revised manuscript.

In revising the manuscript for further consideration, your revisions should address the specific points made by each reviewer and the editors. Regarding comments from reviewer 2, we ask that you include additional covariates in your model/sensitivity analyses to alleviate the concerns noted by the reviewer or clearly outline the assumptions made and their limitations. 

Please also check the guidelines for revised papers at http://journals.plos.org/plosmedicine/s/revising-your-manuscript for any that apply to your paper. In your rebuttal letter you should indicate your response to the reviewers' and editors' comments, the changes you have made in the manuscript, and include either an excerpt of the revised text or the location (eg: page and line number) where each change can be found. Please submit a clean version of the paper as the main article file; a version with changes marked should be uploaded as a marked up manuscript.

We expect to receive your revised manuscript by Nov 16 2020 11:59PM. Please email us (plosmedicine@plos.org) if you have any questions or concerns.

We look forward to receiving your revised manuscript. 

Sincerely,

Adya Misra, PhD

Senior Editor 

PLOS Medicine

plosmedicine.org

Comments from the reviewers:

Reviewer #1: The authors have addressed my points. I still think it is a shame that we do not know whether there is a threshold below which further BP reduction is pointless but their data shows that this is only a minority interest.

Michael Dewey

Reviewer #2: Dear Editors of PLOS Medicine, 

It is with great interest that I read the revised version of "Blood pressure lowering treatment for prevention of cardiovascular events in patients with atrial fibrillation: an individual-participant data meta-analysis" by Prof. Rahimi et al. 

Firstly, I must emphasize that I think this meta-analysis is of huge importance clinically. Before, guidelines have only had the ACITVE-I trial and observational data to lean on, giving advice on BP lowering in atrial fibrillation. Thus, the current BPLTTC analysis will have huge impact, and therefore it needs to be great. For simplicity, I will follow the same structure as in my previous set of comments. 

1) Ok, so it builds on a systematic literature search (not the same as systematic review). This should be reported in the manuscript (or appendix), not only referred to the published protocol. The authors should consider a PRISMA flowchart like the one in the protocol, but with an additional level below the 100 trials eligible for consideration, explaining what they explained in text in their response. Also, the funnel plot they use as an argument against acquisition bias should be added to the appendix. A systematic review needs to be 100% transparent. 

2) I´m very happy that the authors decided to perform risk of bias assessments. However, these are not included in the appendix submitted. It does sound strange, though, that all trials were judged to be at low risk of bias. For example, randomization in the CAPPP trial is known to have failed, as pointed out elegantly by Richard Peto in a letter to the editor few months after the original publication (https://doi.org/10.1016/S0140-6736(05)75340-X). In EWPHE, 36% of participants stopped the double-blind phase early; 15% of participants were lost to follow-up for non-mortality outcomes. The number of participants lost to follow-up widely exceeded the number of participants with cerebrovascular events (all in the original Lancet publication). How do the authors motivate "low risk of bias" in these circumstances? I really hope I´ll get to see the full risk of bias table, including judgements for each domain, in a future submission. 

3) It is very unfortunate that HF data has not been shared since the HF is such an important confounder in BP lowering trials. The authors have done what they can with this comment, although the sensitivity analysis presented in this response must also be added to the appendix. 

4) This is still my main concern, which I seem to be sharing with reviewer 1 (statistical reviewer). Of course, it is possible to include trials without AF participants in the statistical model, but including them will inevitably make AF and non-AF participants in the meta-analysis more different, as I explained in my previous comment. The authors state that this is accounted for, because the model assumes different baseline hazards for each trial, and because treatment arm, trial-level BP differences between treatment arms, and treatment-BP difference interaction, were included as covariates. For such a model to be valid, baseline hazard and BP difference would have to be the only two determinants of treatment effect beyond AF status, under study here. This implicitly assumes that no other trial or patient characteristics are important per se; everything is summarized in the baseline hazard function. This is a quite strong assumption, which, although the authors have argued for it previously, is not agreed upon by all in the hypertension community. For example, Thomopoulos et al (DOI:10.1097/HJH.0000000000001276 ) have found differences in treatment effect between people with and without diabetes, and Brunström et al have found differences between different baseline BP levels in primary prevention, and differences between primary and secondary preventive participants at low baseline BPs (doi:10.1001/jamainternmed.2017.6015). These meta-analyses are of course trial-level analyses, with potential for ecological bias, but nonetheless they are potential arguments for why other things (like DM, baseline BP, and previous CVD), beyond baseline hazard and BP difference, would be important to consider in a model accounting for differences between AF and non-AF participants. 

Based on the difficulties described above, which would be much less severe had non-AF trials been excluded, I think the main analysis should be restricted to trials including both AF and non-AF participants. I completely disagree with the authors´ statement that their one-stage model "did not lead to bias, but improved precision". In my view, that improved precision is false because it comes from trials not including people with the condition under study. This, in turn, is very important because the evidence for AF participants is in fact weaker than suggested by this study. 

If the authors persist including non-AF trials, and the editors chose to accept this (against my recommendation), I would recommend including more covariates (at least the ones mentioned above) in their model to fully account for potential differences in treatment effect between AF and non-AF participants. If the authors persist with the same model, and the editors chose to accept this (against my recommendation), they need to be very explicit about the assumptions that comes with that model, and discuss this under the limitations section. 

5) Again, the argument that population/setting is accounted for by different baseline hazard functions depends on the assumption that no trial or population characteristics interact with the treatment effect. This, quite simplistic, view is also reflected in the argument concerning fixed- and random-effects model. However, I do think that the sensitivity analysis using both types of models is reassuring, as expected when statistical heterogeneity is low. Results from the I2-analyses and the sensitivity analysis should be presented to the readers in the appendix and commented in the article. 

6) Ok, needs to be included in the appendix. 

7) Great, but the conclusion in the main article needs to be toned down as well. "…even when baseline BP is below recommended treatment thresholds" should be removed or rephrased. Although the analysis >/< 140 mm Hg did not find evidence for an interaction, the analysis had low power and the results in each group was not significant on its own. Phrasing in abstract "no evidence that treatment effects varied according to baseline systolic BP or drug class" is more correct. 

One additional minor note is that the Ettehad reference now appears twice. 

Reviewer #3: No additional comments

[LINK]

---

## [Decision Letter · Decision Letter 3]

19 Jan 2021

Dear Dr. Rahimi,

Thank you very much for re-submitting your manuscript "Blood pressure lowering treatment for prevention of cardiovascular events in patients with atrial fibrillation: an individual-participant data meta-analysis" (PMEDICINE-D-20-01625R3) for review by PLOS Medicine.

I have discussed the paper with my colleagues and the academic editor, and it was also seen again by one of the reviewers. I am pleased to say that provided the remaining editorial and production issues are dealt with we are planning to accept the paper for publication in the journal.

[LINK]

We look forward to receiving the revised manuscript by Jan 26 2021 11:59PM.   

Sincerely,

Artur Arikainen, 

Associate Editor 

PLOS Medicine

plosmedicine.org

Requests from Editors:

1. Please address reviewer #2’s final comment regarding HF patients at baseline.

2. Financial Disclosure: Please write in full sentences and also add “The funders had no role in study design, data collection and analysis, decision to publish, or preparation of the manuscript.”; or explain otherwise.

3. Competing Interests: Please update to use our standard text: “KR is an Academic Editor on PLOS Medicine's editorial board."

4. Data Availability Statement: Since the data are not freely available, please describe/reiterate briefly the ethical, legal, or contractual restriction that prevents you from sharing it. Please also include an appropriate contact (web or email address) for inquiries (note: this cannot be a study author).

5. Abstract:

a. Line 29: No need to define BP once more.

b. Please report your abstract according to PRISMA for abstracts, following the PLOS Medicine abstract structure (Background, Methods and Findings, Conclusions) http://www.plosmedicine.org/article/info:doi/10.1371/journal.pmed.1001419. Specifically, please describe how trials were identified/searched for, including database names, search date(s), inclusion criteria, method of quality/bias assessment.

c. Please mention a summary of trial numbers by country/region, and overall study quality/bias.

d. Please include p values, and also quantify your results further for these data: “Meta-regression showed that relative risk reductions were proportional to trial-level intensity of BP lowering in patients with and without AF at baseline.”

e. In the limitations, "difficulty in ascertaining AF cases" could be made more explicit (i.e., did the original investigators have heterogeneous criteria for AF, or did the authors of the present paper have difficulty extracting the information from study reports, or both).

f. Conclusions: Please begin with “In this meta-analysis, we found that…” (or similar)

6. Please move citations to before punctuation, eg: “…[1,2].”

7. Methods: Please mention that separate ethical approval was not required for your study.

8. Results and Figures: Please present p values for comparisons.

9. Lines 346-349: Please delete Funding information from here.

10. Please delete “(London, England)” from the journal name for Lancet references.

11. Please upload each item of Supplementary Information as a separate file, with no tracked changes. Please then include a citation to the supplementary methods in the main Methods section.

12. Please move S1 Figure to be a main Figure.

13. Please report your SR/MA according to the PRISMA guidelines provided at the EQUATOR site: http://www.equator-network.org/reporting-guidelines/prisma/

Please provide the completed PRISMA checklist. When completing the checklist, please use section and paragraph numbers, rather than page numbers. Please also add the following statement, or similar, to the Methods: "This study is reported as per the Preferred Reporting Items for Systematic Reviews and Meta-Analyses (PRISMA) guideline (S1 Checklist)."

----

Comments from Reviewers:

Reviewer #2: Thank you for the opportunity to review this third version of the BPLTTC manuscript on antihypertensive treatment in patients with atrial fibrillation, with Prof Rahimi as the corresponding author. 

Since the previous version, the manuscript has improved significantly. The authors have carefully addressed all my comments. They have extracted additional data (heart failure data from the ACTIVE-I trial), of crucial importance to ascertain the validity of the study findings, and they have performed several additional analyses, reported in the supplement and commented in the main text, with reassuring results. Also, the methods are more transparent, as the supplementary text has been expanded. The authors should be commended for their efforts.

Regarding point 4 in the previous set of comments, where we seem to disagree on the optimal analytical approach, the authors argue for leaving out several covariates of possible interest, because the aim was, quote: "to answer the question of whether a typical patient with AF as included in the trials would benefit similarly from BP lowering treatment as someone without AF. By contrast, the proposed approach by the reviewer is to ensure that patients with AF and no-AF are as similar as possible, which presumably will help understand the independent effect of AF on effect modification". In the methods section, paragraph on statistical analysis, they write, quote "Our main analyses aimed to address four questions: (q1) whether AF status at baseline modifies treatment effects…". Thus, it seems to me we agree that the adjusted analyses would be meaningful to answer the first research question. I see no real problem in using different primary models to address different research questions, and the authors could consider elevating this in relation to q1 in the results. 

One additional comment is that it is stated on row 274 that patients with HF at baseline were excluded from all studies. However, there is also a sensitivity analysis, excluding HF patients from ACTIVE-I. This should be clarified. 

Finally, I would like to congratulate the authors on an interesting and important paper.

----

[LINK]

---

## [Editor Report · Decision Letter 4]

24 Feb 2021

Dear Dr. Rahimi,

Thank you very much for re-submitting your manuscript "Blood pressure lowering treatment for prevention of cardiovascular events in patients with atrial fibrillation: an individual-participant data meta-analysis" (PMEDICINE-D-20-01625R4) for review by PLOS Medicine.

I apologize for the delay responding to your recent inquiry, and thank you for submitting the revised version addressing the editor and reviewer comments. There are a few remaining editorial issues to address, but provided these are resolved and production issues are dealt with we are planning to accept the paper for publication in the journal.

[LINK]

We look forward to receiving the revised manuscript by Mar 03 2021 11:59PM.   

Sincerely,

Caitlin Moyer, Ph.D.

Associate Editor 

PLOS Medicine

plosmedicine.org

Requests from Editors:

1. Similarity with published abstract: It appears that there is some overlap between the content of the manuscript and your published abstract.

Regarding the text overlap, please slightly reword any directly replicated text, if possible, to avoid copyright issues.

For example:

In the abstract: "The hazard ratio for major cardiovascular events was 0.91 in patients with AF (95% confidence interval 0.83 to 1.00) and 0.91 without AF (95% confidence interval 0.88 to 0.93) for each 5-mmHg reduction in systolic BP, with no difference between subgroups. In patients with AF, there was no evidence that treatment effects varied according to baseline systolic BP or drug class."

The final sentence of the Discussion/Conclusion: "Guidelines should be updated to give clear recommendations for pharmacological BP lowering for prevention of cardiovascular events in patients with AF."

In addition, Figure 4 is reproduced from the published abstract: (https://doi.org/10.1093/ehjci/ehaa946.0672). If you would like to include the figure in the manuscript, please contact OUP to obtain their agreement to it being published under our license. An alternative would be to remove the figure and present the data in a different format (e.g. as a table).

2. Data Availability Statement: Thank you for your response "We have now added the data availability statement as requested and explained how the interested authors could ask for BPLTTC dataset." Please update the data availability statement of the manuscript submission form with the information provided on Page 16 Lines 363-366.

3. Please remove the highlighting in the final "clean" version submitted.

4. Methods: Page 6: Lines 126-127: Please provide a reference to the supporting information file for the protocol (e.g. S1_Protocol).

5. Discussion: Line 302: Please revise this sentence, as there may be a word missing here: "...ideal SBP target for prevention of cardiovascular was found..."

6. Page 16: Please remove the Financial Disclosure and Data Availability Statement sections from the main text, and ensure the information is entered in the appropriate section of the manuscript submission form.

7. References: Please double check the formatting of references, using NLM journal title abbreviations where appropriate (e.g. JAMA Intern Med instead of JAMA Internal Medicine). Please use the "Vancouver" style for reference formatting, and see our website for other reference guidelines: https://journals.plos.org/plosmedicine/s/submission-guidelines#loc-references

[LINK]

---

## [Editor Report · Decision Letter 5]

25 Mar 2021

Dear Dr Rahimi, 

On behalf of my colleagues and the Academic Editor, Gregory Lip, I am pleased to inform you that we have agreed to publish your manuscript "Blood pressure lowering treatment for prevention of cardiovascular events in patients with atrial fibrillation: an individual-participant data meta-analysis" (PMEDICINE-D-20-01625R5) in PLOS Medicine.

PRESS

Sincerely, 

Caitlin Moyer, Ph.D. 

Associate Editor 

PLOS Medicine